# Responsive and Minimalist App Based on Explainable AI to Assess Palliative Care Needs during Bedside Consultations on Older Patients

**Vicent Blanes-Selva** [1,*] **, Ascensión Doñate-Martínez** [2] **, Gordon Linklater** [3] **, Jorge Garcés-Ferrer** [2] **and Juan M. García-Gómez** [1]

1   Biomedical Data Science Laboratory, Instituto Universitario de Tecnologías de la Información y Comunicaciones (ITACA), Universitat Politècnica de València, 46022 Valencia, Spain; juanmig@ibime.upv.es
2   Polibienestar Research Institute, University of Valencia, 46022 Valencia, Spain; ascension.donate@uv.es (A.D.-M.); jordi.garces@uv.es (J.G.-F.)
3   Highland Hospice and NHS Highland, Inverness IV2 3BW, UK; Gordon.Linklater@nhs.scot
*   Correspondence: viblasel@upv.es

**Abstract:** Palliative care is an alternative to standard care for gravely ill patients that has demonstrated many clinical benefits in cost-effective interventions. It is expected to grow in demand soon, so it is necessary to detect those patients who may benefit from these programs using a personalised objective criterion at the correct time. Our goal was to develop a responsive and minimalist web application embedding a 1-year mortality explainable predictive model to assess palliative care at bedside consultation. A 1-year mortality predictive model has been trained. We ranked the input variables and evaluated models with an increasing number of variables. We selected the model with the seven most relevant variables. Finally, we created a responsive, minimalist and explainable app to support bedside decision making for older palliative care. The selected variables are age, medication, Charlson, Barthel, urea, RDW-SD and metastatic tumour. The predictive model achieved an AUC ROC of 0.83 [CI: 0.82, 0.84]. A Shapley value graph was used for explainability. The app allows identifying patients in need of palliative care using the bad prognosis criterion, which can be a useful, easy and quick tool to support healthcare professionals in obtaining a fast recommendation in order to allocate health resources efficiently.

**Keywords:** palliative care; assessment; mortality; webapp; bedside; machine learning

## 1. Introduction

The World Health Organization (WHO) [1] defines healthcare sustainability as "the ability to meet the needs of the present without compromising the ability to meet future needs". This topic is increasingly important [2] and possibly especially challenging in developed countries, where ageing raises healthcare costs [3]. However, regardless of the health policies in different countries, there is an interest in cost-effective alternatives that deliver (at least) the same quality of care, especially among older populations and those with chronic conditions and morbidity. Palliative care (PC) has gained the attention of clinicians and researchers in recent years as both an alternative to standard care and through a combined approach for these patient groups [4].

According to the recent redefinition of PC by Radbruch et al. in 2020 [5]: "Palliative care is the active holistic care of individuals across all ages with serious health-related suffering due to severe illness and especially of those near the end of life. It aims to improve the quality of life of patients, their families and their caregivers". Several studies have shown positive clinical results in patients involved in PC programs: an improvement in the quality of life and mood [6,7], symptom control [8] and the reduction of emergency department visits and hospitalisations [9].

Besides the clinical implications, the economic impact of PC programs has also been studied. In 2010, Simoens et al. [10] reviewed different studies trying to compare the cost of PC against standard care; the authors found that PC was consistently cheaper. Kyeremanteng et al., in 2016 [3], reviewed how PC affected the length of stay in the intensive care unit (ICU), which is an expensive form of healthcare. The authors found that PC consultations tend to reduce the length of stay in the ICU. Similarly, Smith et al. [11] found in another review that PC programs are frequently less expensive than comparator groups, and, in most cases, the difference is statistically significant.

It is estimated that approximately 75% of patients nearing end-of-life may benefit from PC interventions [12]. The same authors projected an increase of 25% to 42.4% of people requiring PC in England and Wales, and it is reasonable to expect other developed countries will also have increased PC needs. However, despite the sound evidence of PC being clinically beneficial and cost-effective, it remains under-resourced in many countries.

Clearly, it is important to identify those patients who may benefit from PC at the appropriate time, since those interventions are positive in clinical and economic healthcare areas. Different criteria have been used, also known as "triggers", based on different clinical diagnoses or the detection of more personalised PC needs [13]. A limited prognosis is still a widely used criterion when screening for patients who may benefit from PC interventions [12]. The surprise question (SQ) has been widely used and promoted to identify patients likely to die within the next year, and therefore possibly benefit from PC [14]. However, the SQ performs poorly to modestly, and further studies are needed to develop more accurate tools [14]. In addition to the SQ, there are other tools aimed to predict all-cause mortality. Some of the more accurate ones are data-driven and based on statistics and machine learning: the PROFUND index [15], HOMR [16], which has been modified in further publications [17,18], or a deep learning approximation [19]. In addition, our team presented a machine-learning-based approximation that targets adults ($\geq$18) [20].

Despite the different literature approximations to this problem and the good predictive power reported for the different models, implementing these predictive models in clinical practice is not easy. The clinical decision support system (CDSS) implementation in one organisation can fail due to user acceptance, as potential users can consider the outputs irrelevant or unreliable, or that the CDSS interfere in their workflow [21]. Leslie et al. [22] provide a list of key features that are very important during the CDSS design, which we can summarise as follows: the tool should meet the user needs, adapt to the clinical workflow and be flexible enough to allow the healthcare professional to manage in their own way.

In concrete, CDSS powered by machine-learning technology has been utilised successfully in several clinical applications, especially dealing with medical imaging in fields such as radiology, dermatology or ophthalmology [23]. However, other fields, such as biomarker discovery or clinical outcome prediction, have also benefitted from incorporating machine learning algorithms. In the review performed by Yu et al. [23], two categories of challenges are discussed: technical challenges, which includes the data quality and interpretability of the models, and social, economic and legal challenges, where the author remarks on the importance of the integration of the tool with the clinical workflows.

A particularly interesting use of CDSS is in bedside applications, where the clinical professionals do not need to spend much time working on documentation or on a desktop computer using the available tools through mobile devices [24]. Some studies focused on the usability of mobile clinical decision support models have proposed checklists, also known as heuristics, to determine if applications comply with usability standards. A recent example is Reeder et al. [25], who compile a heuristic focused on mobile CDSS applications. When designing these bedside apps, one essential principle is to show only the necessary information needed to proceed. In other words, the application should be as minimalist as possible, whilst keeping the original functionalities and objectives.

Some approximations to bedside mortality risk assessment have been studied in the past, proposing scores and indexes for different health issues, such as acute pancreatitis [26], acute ischemic stroke [27], non-cardiac surgery [28] and, recently, for COVID-19 [29]. Despite

the increase in research around predictive models using mobile applications in the health domain, we found very little information about any apps used in practice. One of the few apps available is the COVID-19 mortality calculator [29]. However, to the best of the authors' knowledge, there is no mobile app to predict 1-year general mortality at the bedside.

The aim of this study is twofold. On the one hand, the aim is to create a compact version of the 1-year mortality model using common and easy to gather variables during admission based on the larger model developed by the authors previously [30]. On the other hand, the aim is to implement a web application (web app) so that health care professionals can assess PC needs during bedside examinations utilising a smartphone or a tablet.

## 2. Materials and Methods

### 2.1. Ethics

The data used in this study came from the University and Polytechnic La Fe Hospital of Valencia and were retrospectively collected on the electronic health records (EHR). This procedure was assessed and approved by the Ethical Committee of the University and Polytechnic La Fe Hospital of Valencia (registration number: 2019-88-1).

### 2.2. Data

This study makes use of the same dataset as the one described in [30]. EHR information was collected from admissions for older patients ($\geq$65 years old), excluding those admitted to the psychiatry department. Data comprise 1 January 2011 to 31 December 2018, containing 39,310 different admission episodes from 19,753 different patients. Patients' death date was available in the EHR and was used to calculate the one-year mortality target variable.

### 2.3. Feature Selection and Modelling

To create a compacter model, we applied a strategy to reduce the number of variables, starting with the list of the 20 most important variables obtained from our previous work [30]. In addition, we removed the administrative variables that may be not compatible with other information systems: diagnosis-related group (DRG), admission code and department code. The final list of variables composing the dataset is available in Table 1.

**Table 1.** Variable summary and ranking.

| Variable | Rank | Mean $\pm$ Std | Missings |
|---|---|---|---|
| Number of Active Groups (Meds) | 1 | 2.44 $\pm$ 3.68 | 0% |
| Charlson Index | 2 | 4.77 $\pm$ 3.34 | 0.2% |
| Barthel Index | 3 | 51.91 $\pm$ 39.75 | 73.4% |
| Metastatic Tumour | 4 | -[1] | 0% |
| Age | 5 | 79.4 $\pm$ 8.36 | 0% |
| Urea (mg/dL) | 6 | 61.19 $\pm$ 43.36 | 37.1% |
| RDW SD [2] (fL) | 7 | 49.66 $\pm$ 7.36 | 21.7% |
| Leukocyte ($10^3$/$\mu$L) | 8 | 9.23 $\pm$ 6.85 | 21.6% |
| RDW CV [3] (%) | 9 | 15.26 $\pm$ 2.43 | 21.7% |
| Sodium (mEq/L) | 10 | 139.89 $\pm$ 4.92 | 21.2% |
| C Reactive Protein (mg/L) | 11 | 55.1 $\pm$ 70.54 | 47.3% |
| Creatinine (mg/dL) | 12 | 1.23 $\pm$ 1 | 20.7% |
| Haematocrit (%) | 13 | 36.24 $\pm$ 5.8 | 21.6% |
| Glucose (mg/dL) | 14 | 122.5 $\pm$ 54.98 | 24.0% |
| Number of Previous ER | 15 | 6.04 $\pm$ 6.56 | 1.4% |
| Number of Previous Admissions | 16 | 7.62 $\pm$ 7.49 | 0% |
| Potassium (mEq/L) | 17 | 4.22 $\pm$ 0.61 | 22.5% |

Table 1 legend: [1] Distribution for categorical variable metastatic tumour is Yes: 77 (0.2%); No: 38339 (99.8%). [2] RDW SD = red blood cell distribution width standard deviation. [3] RDW CV = red blood cell distribution width coefficient of variation.

In the next step, we ranked the variables according to their importance using an iterative method. The algorithm starts with a whole list of variables, and then an explainable model is trained using them and the importance of every variable is obtained. We used the random forest model as this explainable model, using the GINI criteria to determine the importance of each variable after the model was fitted. After this, the variable with less importance was removed from the list. The process ended when only one variable was left. The ranking of every variable was the iteration when it was removed from the list in reverse order.

To determine the optimum number of variables, we took the ranking obtained in the previous step and applied the following algorithm: starting with the most relevant variable, a model-based gradient boosting machine is trained and validated with the 10-fold validation method, using nine sections of the data to train the model and one to test it. The area under the ROC curve (AUC ROC) [31] is computed for the 10 test splits, averaged and stored as a result for the first variable. The process continues, adding the following variable in the ranking to the model until the final iteration, where all variables are included in the model.

Among the selected variables (Table 1), there are missing values, except in the age and the number of active groups. We hypothesised that the main mechanism producing these missing values is the clinical criterion, where tests are not performed if the physicians do not consider their results important to diagnose or treat the patient. The missing values produced by this mechanism are known as missing not at random (MNAR) [32]. To use the maximum amount of information, an imputation method was needed. In our case, we combined the inclusion of an imputation mask (a dummy variable indicating if the data are present) and an iterative imputation technique [33]. We decided to use both techniques following the results of Sperrin and Martin [34] because it improves the modelling when dealing with MNAR, but has no detrimental effects for missing at random. Following the original data criteria, the only required input for the user is the age and the number of active groups prescribed to the patients during admission.

### 2.4. Explainability Layer

When a new sample introduced by the user reaches the gradient boosting machine, an explainer object is created to interpret the effect of the different variables of the sample in the prediction. This explainer object is the TreeExplainer, which is implemented in the SHAP library [35]. The output of this process is the Shapley values. Every value corresponds to one variable; positive values indicate that the value on this variable pushes the prediction to the positive class, which is mortality within the year in our case. In addition, the bigger the absolute value, the greater the effect on the prediction. We have created a bar graph to represent these values visually, showing only the most relevant ones (bigger absolute value) and adopting the green for the negative values (greater for the patient survival) and red for the positive ones.

### 2.5. APP Implementation and Software

Finally, and after determining the optimal number of variables, a model was trained using all the available cases. The model was then deployed in a publicly available Django web application. We used the Bootstrap library to make the website responsive and adapt the size and elements display in all screen sizes. The app's interface was designed to be as minimalist and functional as possible, following the heuristic from Reeder et al. [25] when applicable. Figure 1 illustrates the whole methodology workflow. Model creation and evaluation process were performed using the NumPy [36], pandas [37] and scikit-learn [38] libraries working with the Python programming language in its 3.8 version.

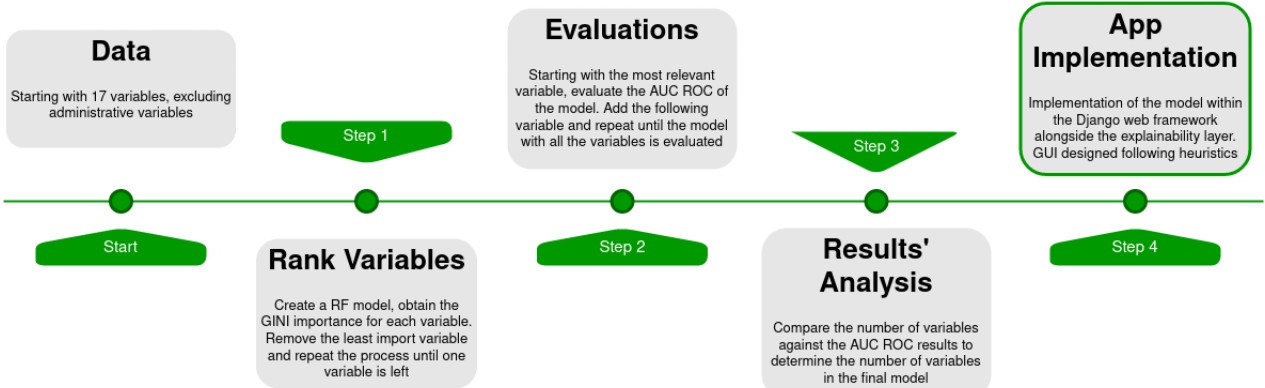

**Figure 1.** Methodology process summary from the initial data to the web application.

## 3. Results

Table 1 shows the results of the data exploration through variable parameters and the number of missing values. In addition, it includes the importance ranking obtained for each variable. The most relevant variables are the number of active groups (in the medications provided), the Charlson Index, the Barthel Index and the patient's age.

Results for the second experiment can be observed in Figure 2. Each coloured bar represents the mean AUC ROC using the 10-fold model with the N most important variables determined by the previous experiment. Grey bars on the top represent the 95% confidence interval. The mean AUC ROC values increase as the number of variables grow, but changes are not statistically significant when the number of variables is greater than nine.

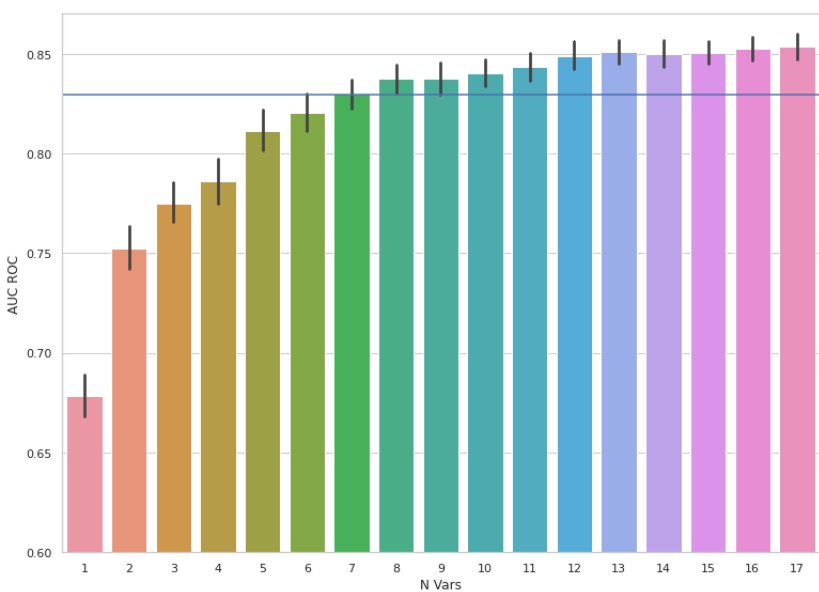

**Figure 2.** AUC ROC values for each number of variables. The horizontal line marks the result for the selected number of variables in the final version.

Figure 3a,b show the final aspect of the tool when accessing it from a smartphone. Figure 3a presents the form where the health care professional enters the patients' data. Units have been added to laboratory tests fields, as well as a tooltip with the reference values to help the users. The number of variables has been selected according to the heuristic principle of "Is only (and all) information essential to decision making displayed on the screen?" and "Has the need to scroll been minimized and where necessary, are navigation facilities repeated at the bottom of the screen?". The order and position of the layout have been influenced as well by the heuristic. Figure 3b shows a results example,

a modal panel that appears over the previous screen, including the numeric value for the prediction in percentage, and the variable importance bar plot built from the Shapley values. The missing indicators are displayed on the bar graph with the suffix "missing" if the explainer found the variable relevant.

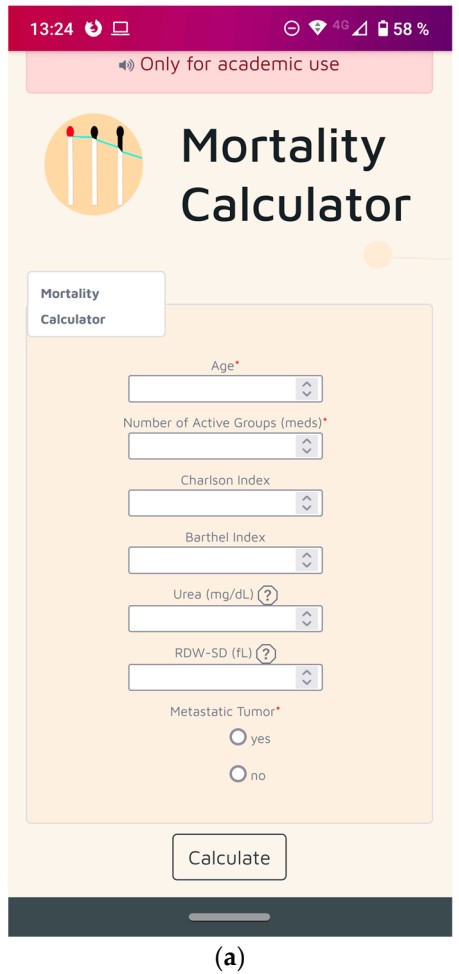

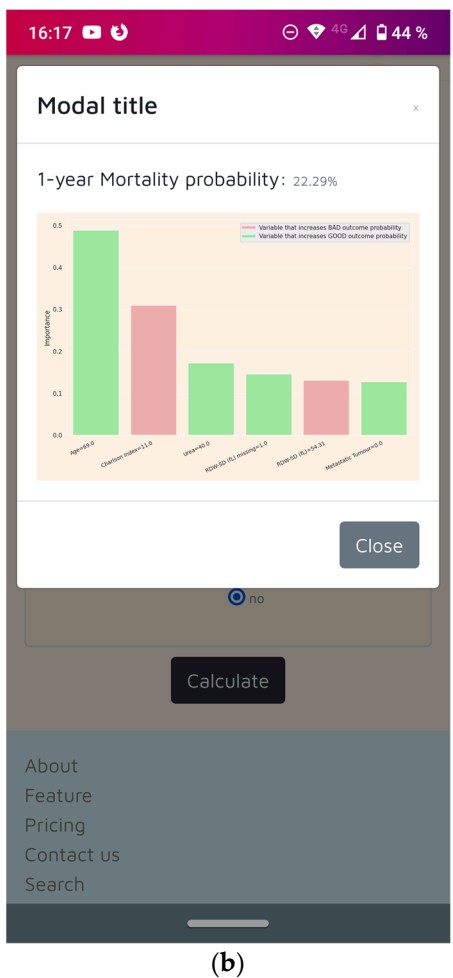

(**a**)    (**b**)

**Figure 3.** Screenshots from the application running on Firefox Focus in Android device: (**a**) main screen where the user has to input the data; (**b**) results and explainability graph.

## 4. Discussion

The experimentation reported in the first place which variables were more informative to create the predictive model. With this information, we calculated the metrics of the model using an increasing number of variables to determine the optimum number of variables. Our objective was to maximise the performance of the model while keeping the required number of inputs low. Setting a low number of required inputs is important from the design perspective [39] to minimise the time required to use the tool, which directly affects the interference in the clinical workflow and the chances of successful implementation [22]. Therefore, we followed the heuristic by [25] when possible, although some checkpoints did not apply due to the app's small size.

Using the minimalist approach, the authors believe that seven variables is the most convenient number to incorporate in this model. This supposes that the final evaluation of the model is an AUC ROC of 0.83 with a 95% confidence interval of (0.82 to 0.84). Arguments favouring increasing or reducing the number of variables can be made, but there is no precise study about how this factor affects repetitive tasks over time. Previous mortality models proposed in the literature have achieved a greater predictive performance,

such as HOMR (0.89 to 0.92 AUC ROC) in its different versions [16–18], Avati's deep learning approach (0.87 AUC ROC on admitted patients) [19] and our previous work, which achieved 0.91 of AUC ROC [20].

However, this is not a fair comparison for three main reasons. First, the previously mentioned studies are aimed at adults ($\geq$18), and Avati's work even includes paediatric information, whereas this work was focused on older patients ($\geq$65). Since age is a determinant mortality factor recognised in the literature [15–20], limiting the dataset to older patients removes younger and healthier patients and makes classification more difficult. There are few 1-year mortality admission predictive indices focused on older patients. Inouye et al. [40] reported an AUC ROC of 0.83 in development and 0.77 in validation, whereas Fischer et al. [41] reported 0.82 in development. Both studies report a similar performance to our results. Still, the indices were developed with a relatively small number of cases, 525 and 435, respectively, which may lead to weaker results than modern studies with bigger samples. Secondly, despite some of these studies using easily obtained variables, such as the HOMR-now! [17] derived from the HOMR model using data immediately available at hospital admission, none of them have been designed to be a compact and bedside clinical decision support tool. Lastly, with the design of this model, administrative variables have been avoided to predict mortality as they could be difficult to obtain or could not be exchangeable between different healthcare systems, such as the department where the patients are admitted. Other works in the literature include variables that may present difficulties implementing the system in different hospitals due to having varying protocols or administration; for example, the original HOMR study [16] used, as variables, if the admission has been performed by ambulance or if the patient receives oxygen at home.

The variables that compose this predictive model are similar to those used in other studies. The Charlson Index was also used in HOMR and HOMR-now! [16,17]; The Barthel Index had been used in the PROFUND index [15]; blood urea nitrogen was the second most relevant variable in our previous study [20] and some studies have identified that this variable is associated with mortality in different contexts [42–44]. Age has been used in all of the previously mentioned models, and is strongly associated with mortality. Other variables have been tested in their association with mortality. In our study, the number of active groups in medication resulted in one of the most informative predictors, despite other studies not finding a direct association [45]. Gelder et al. [46] considered this variable in their 90-days mortality model, but it did not make the cut of the final model. Using the explainability method, we detected a strong negative interaction when the variable has a value of 0. We hypothesise that this is the case for terminal patients on non-curative treatment. Lower values for this variable have positive effects on the prediction, whereas this effect declines and turns negative as the value increases. This pattern is expected, since patients with more prescribed medications are likely to have a more complex health status and an increased number of comorbidities. RDW-SD measures the standard deviation in the variability of red cell volume/size (RDW). In 2010 Patel et al. [47] conducted a meta-analysis where the RDW test was a strong predictor for mortality in older adults. The metastatic tumour is a condition related to survival and has been used in previous works as well [20].

Accurate predictions can be helpful to health care professionals using these types of tools, but the users need to understand why a specific prediction has been made. As Shortliffe and Sepúlveda stated in [48], black boxes are unacceptable. It is necessary to provide an explanation of why the tool has made a certain recommendation in order for the professionals to accept or override it. Carroll et al. [49] also point to the lack of explanation for the proposed solution as a pitfall for CDSS. In this sense, in this work, we tried to address this situation by providing a graphic representation of the Shapley values [38]; this graphic shows the most relevant variables pushing the prediction to higher or lower mortality within the year probability.

Prognosis-based interventions are one of the most common ways to detect patients in need of PC. According to Kayastha and Leblanc [13], there are two other identification types: need-based and trigger-based. On the one hand, need-based assessments use different tools, such as questionnaires and the need from the clinician's agreement on how to interpret the results, which could lead to too many referrals. On the other hand, trigger-based assessments define a set of conditions to trigger the intervention that is a mix of both prognosis and needs. The tool proposed in this work falls into the prognosis-based category. One of the downsides of this approximation is that it does not provide information about the concrete needs a patient may have. It may be interesting to complement the output provided from our tool with other criteria in order to apply an additional filter and improve the PC needs identification. Recently, Wegier et al. [50] studied this possibility in their work by assessing PC needs using symptomatology and readiness to engage in advance care planning on patients already identified using their predictive model mHOMR. Most of the patients identified presented unmet PC needs in accordance with the study criteria. This result opens the possibility to improve the effectiveness of mortality models with other PC needs assessment tools.

Resource allocation to PC programs is a very relevant challenge because these programs present a list of clinical benefits to the patients but are often under-resourced [6–9]. If available, PC is usually provided to patients with a high symptom burden or in the terminal phase of illness. However, a shift to an early PC approach is advocated [51]. In addition to this, some studies have proven different PC interventions to be cost-effective compared to standard care [3,10,11]. Early PC has shown lower hospital costs during hospital admission [52] or costs savings over routine care [53]. Combined with the anticipated increased PC demand alongside limited resources, these facts create the necessity to identify which patients could benefit more from PC interventions. A bedside tool, such as the one presented in this work, can help identify those patients in an agile way, so that the PC delivery can be improved and, therefore, bring clinical benefits and less expensive care to those in need, improving the sustainability of the healthcare system.

The main goal of this work was to create a quick-to-use tool to determine which patients may benefit from PC. Besides the identification power, the main benefit of having a compact tool is the reduced time needed for completion, which is essential to the tool's success [48,49]. The main strengths of this study are, first, the creation of a compact all-cause mortality model during hospital admission, obtaining a good discriminative power of 0.82 of AUC ROC. Second, this minimalist design, in addition to the lack of hospital-specific variables, allowed us to create a clinical decision tool as a web app to be used in any portable device with an internet connection and a web browser. Finally, the tool provides a numerical result and uses explanatory techniques to help the health care professionals in their final decision making, providing more context to accept or override the prediction. This simple tool presented a good predictive power and can quickly detect patients in need of PC and aid the effective allocation of resources to improve healthcare sustainability.

The main limitation of this study resides in its validation. Evaluation of the predictive model has been performed only within the same dataset using the K-fold validation strategy from a single institution. Thus, we cannot ensure the reported effectiveness in other contexts or other populations [54]. External validation with other hospitals and populations is needed. Another important limitation of this tool is that the data input and the output calculation are disconnected from the EHR system. Simplifying the data input for the models to be used in a bedside tool was addressed during the study methodology. Still, the output prediction is not recorded anywhere, so it will require a manual introduction of the results in another system; this could prove difficult for both the case review process for healthcare workers and for the acceptance of the tool for other stakeholders, such as hospital administrators or policymakers, who are unable to obtain a 'big picture' from the application records [55,56]. However, the disconnection from the EHR and the application not storing any data solve most of the privacy and data security difficulties, which are a barrier to the large scale adoption of mHealth applications [55].

## 5. Conclusions

Older populations with chronic conditions or multimorbidity are increasing, which may mean an increased demand and use of health care services, with PC interventions among them. Unfortunately, there are many barriers in accessing PC, such as limited resources or late referrals when a person is in their last phase of the end-of-life process. This minimalist and simple predictive model can support the early identification of patients in need of PC, requiring only a minimum investment of time by clinicians. Tools such as this can facilitate the management of complex patients and overcome decision-making difficulties in integrating PC in daily clinical practice. A better PC identification delivers clinical benefits to the patients and could help to allocate resources, improving the system's sustainability. The tool is available at: http://palliative-calculator.upv.es/ (accessed on 29 August 2021).

**Author Contributions:** Conceptualization, V.B.-S. and J.M.G.-G.; experimentation and development, V.B.-S. and J.M.G.-G.; manuscript writing, V.B.-S., A.D.-M., G.L. and J.G.-F.; critical review A.D.-M., G.L., J.G.-F. and J.M.G.-G. All authors have read and agreed to the published version of the manuscript.

**Funding:** This work was supported by the InAdvance project (H2020-SC1-BHC-2018–2020 grant agreement number 825750.) and the CANCERLEss project (H2020-SC1-2020-Single-Stage-RTD grant agreement number 965351), both funded by the European Union's Horizon 2020 research and innovation programme.

**Informed Consent Statement:** Not applicable.

**Data Availability Statement:** Not applicable.

**Acknowledgments:** The authors thank their contributions to María Soledad Giménez-Campos, María Eugenia Gas-López, María José Caballero Mateos and Bernardo Valdivieso. Special thanks to Ángel Sánchez-García for his contributions to the website.

**Conflicts of Interest:** The authors declare no conflict of interest.

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
