# Peer review of "Responsive and Minimalist App Based on Explainable AI to Assess Palliative Care Needs during Bedside Consultations on Older Patients"

_sustainability, doi:10.3390/su13179844_

Round 1

Reviewer 1 Report

this paper could benefit of a reshaping and shortening of its background paragraph, I would consider as principal aim the second one proposed. I would discuss more about strength and limitations of this solution.
Some terms used seems unlikely in an English paper,  please recheck and proofread 

Reviewer 2 Report

The manuscript described the development of a predictive model and GUI APP for assisting the application of palliative care.  Overall, it scientifically sounds well.  

  1. I suggest to add a flowchart to schematically elaborate the training process of the prediction model and the structure of its GUI APP, so that readers can understand the design of the whole system with ease.
  2. As the most predictive variable is “Number of Active Groups”, it is better to additionally discuss the relationship between “Number of Active Groups” and “Mortality”.
  3. In desktop version of the web APP, the layout of the calculation result is not well.  The text on the x-axis cannot be displayed correctly.  Also, the text size can be enlarged.  I used Google Chrome Version 92.0.4515.131 (Official Build) (64-bit) to test it.
  4. It will be more informative for user if the developer of the APP can additionally put a “question mark” right next to the six variables allowing users to click on it. The explanation for the referred variable will be then pop up.

Reviewer 3 Report

The creation and validation of a bedside web application for palliative care is timely in healthcare. But I have few comments that needs to be addressed prior to further consideration:

  1. As the authors were implementing an app, yet validating its performance for the first time, it is essential to conceptualize the conventional methods used in healthcare settings to determine the routine palliative care needs to the navigation using AI technologies. Gaps in conventional practice need to be addressed and subsequently the conceptualization on the adoption of AI technologies (machine learning and deep learning functions and algorithms) is fundamental to be described in the introduction part. I recommend authors to have a subsection under introduction part on description of AI technologies and its adoption to clinical practice.
  2. Also, I recommend authors to use some AI terms rather than conventional terms – this include: dependent variable (change to AI term – “label or class”); independent/predictor variable (change to AI term – “feature”). Also, the MI used was supervised or unsupervised learning?
  3. In the methods part, it will be essential to report a sub-section on “Statistical analyses,” how was it done and software used for analysis.
  4. Results - Table 1 – It is unclear how were variable ranks determined?
  5. Table 1 – the explanation of variables at the title should be moved to the footnote. Also, what does (*1) mean in the metastatic tumor row?
  6. Please check in Table 1 as well, some of the reported SDs are larger than the mean.
  7. The limitation part should be expanded, especially from the context of clinical setting and accessibility via portable apps such as smartphones or tablets. The following should be addressed:
  • Implications for patients (privacy concerns)
  • Implications for healthcare workers (concerns of interoperability)
  • Implications for stakeholders (issues of acceptance, value for money, safety and confidentiality.

The following literature could be referred:

Petersen C, Adams SA, DeMuro PR. mHealth: Don't Forget All the Stakeholders in the Business Case. Med 2 0. 2015;4(2):e4. Published 2015 Dec 31. doi:10.2196/med20.4349

Kurubaran Ganasegeran, Surajudeen Abiola Abdulrahman. Chapter 3: Adopting m-Health in Clinical Practice: A Boon or a Bane? In: Hemanth JD & Balas VE (Eds). Telemedicine Technologies: Big Data, Deep Learning, Robotics, Mobile and Remote Applications for Global Healthcare. Elsevier, Academic Press (United States). 2019; 1: 31-42.

Round 2

Reviewer 3 Report

Thank you for your revisions.